# The Identification and Driving Factor Analysis of Ecological-Economi Spatial Conflict in Nanjing Metropolitan Area Based on Remote Sensing Data

Ji Cao [1], Weidong Cao [1,*], Xianwei Fang [1], Jinji Ma [1], Diana Mok [2] and Yisong Xie [3]

1   School of Geography and Tourism, Anhui Normal University, Wuhu 241002, China
2   Department of Management and Organizational Studies, The University of Western Ontario, London, ON N6A 5C2, Canada
3   Environment Protection Key Laboratory of Satellite Remote Sensing, Aerospace Information Research Institute, Chinese Academy of Sciences, Beijing 100094, China
*   Correspondence: weidong1@ahnu.edu.cn

**Abstract:** The rapid socio-economic development of the metropolitan area has led to the continuous deterioration of the ecological environment. This leads to intense competition and conflict between different spatial use types. Spatial conflict research is essential to achieve ecological-economic coordination and high-quality development. However, existing studies lack comprehensive and direct ecological-economic spatial conflicts, especially those on the spatial-temporal evolution and potential drivers of spatial conflict. In this study, we identified the ecological-economic spatial conflicts in the Nanjing metropolitan area in 2010, 2015, and 2020. This study used the random forest to analyze the factors that influenced the change of spatial conflict. Results show that: (1) From 2010 to 2020, the ecological-economic spatial conflict in the Nanjing metropolitan area changed significantly. (2) Land use change has an important effect on spatial conflicts, which are easily triggered by uncontrolled urban expansion, but ecological land can mitigate spatial conflicts. (3) Relevant driving factors of spatial conflicts show multi-level features, so the development of conflict reconciliation countermeasures needs to be tailored to local conditions. This study provides a significant foundation for the high-quality development of the Nanjing metropolitan area and provides a reference for the planning and management of the territorial space.

**Keywords:** spatial conflict; conflict identification; analysis of driving factor; the Nanjing metropolitan area

## 1. Introduction

In sustainable development, the development of human society should not only meet the needs of contemporary people, but also should not damage the ability of future generations to meet their own needs. "Space" is the collective term for the Earth's surface area that is suitable for human economic and social activities [1]. As an objective geographical phenomenon caused by spatial resource scarcity and spatial function overflow, spatial conflict is caused by the competition for spatial resources in human activities [2]. Although there are relatively few studies related to spatial conflicts, scholars have noticed the widespread phenomenon of spatial conflicts in society and have put forward the concepts of "regional deprivation" and "spatial competition", which are similar to spatial conflicts. Those concepts' connotations are all contradictory phenomena in the process of spatial conflict. For example, the collision between urban space and rural space, the encroachment of arable land by urban expansion, the competition between green ecological space and urban construction land, and the decline of ecological environment quality due to the disorderly expansion of urban construction land are all important manifestations of the phenomenon of spatial conflict. Along with the continuous development of the social

economy, human activities now have far more influence on the Earth's surface than any period in history. The friction generated by human activity space and natural ecological space also exceeds the limit that the local environment can bear. The degradation of the ecological environment caused by rapid urban expansion makes the ecological-economic space conflict increasingly fierce [3]. The ecological-economic space conflict has also become an important reason hindering sustainable development, which has attracted the continuous attention of scholars [4].

Previous studies have provided various identification and evaluation methods of ecological-economic space conflicts, but most existing studies on ecological-economic conflict are based on panel data and lack spatial analysis. In the past few decades, rapid advances in remote sensing (RS) and geographical information systems (GIS) technology have provided the basis for spatial data acquisition and analysis, which provides an accurate source of data for monitoring and detecting land use, ecological changes, and human activity intensity [5,6]. The spatial data provided by remote sensing can reflect the spatial variation in regions better than traditional statistical surveys. As the carrier of ecological environment and economic activities, land use conflict is the most direct manifestation of ecological-economic spatial conflict and the earliest research involving ecological-economic space conflict [7]. Scholars determine the types of space they belong to according to different land use patterns, and they have found that potential spatial conflicts may arise from different land use patterns overlapping in space [8–10]. Based on the actual situation in China, Chinese scholars have put forward the theory of "ecology-production-life" to study spatial conflict, which is based on the three pillar theories of sustainable development. In the ecological-production-life framework, space is divided into three independent and interrelated spaces: ecological, production, and living space, which makes it easier to distinguish conflicts between different spaces [11]. Meanwhile, this theory also coordinates the relationship between ecological, production, and living space and offers countermeasures and suggestions for the sustainable development of society [12].

In addition, domestic and foreign scholars also analyze the ecological-economic space conflict from the perspective of landscape [13]. The spatial conflict caused by the unreasonable spatial structure of land use can be accurately identified by the landscape pattern theory [14,15]. Scholars use landscape pattern theory to build a spatial conflict model to describe the impact of human activity or natural environment change on the landscape composition, structure, and function and also use relevant landscape pattern models to measure spatial conflict [16,17]. As attention to ecological protection has increased, a conflict analysis method that can directly link ecology and the economy has also made great progress [18–20]. The theory of ecosystem service is considered the bridge between natural environmental systems and social-economic systems [21]. Based on the "land use mode-ecological process-ecological service system" [22], the contradiction between regional ecological and economic space can be directly analyzed through the contradiction between the supply and demand of ecosystem service theory [23–26].

In a word, the existing studies can basically reflect the ecological-economic space conflict, but they are mostly specific ecological aspects, such as for wetlands, water, protected natural areas, and other social and economic space conflicts, and there is still a lack of overall analysis of the ecological-economic space conflict [27–29]. Moreover, the identification of spatial conflicts relies too much on land use change, neglecting other socio-economic and natural environmental factors. The existing related methods also have their disadvantages. The "ecology-production-life" theory divides space into three separate spaces, thus separating the integrity of space, and cannot well analyze the space conflict in complex situations. Due to its theoretical characteristics, the landscape pattern theory considers economic factors less [30]. Influenced by dynamic changes in time and space, the supply and demand studies of ecosystem services are greatly influenced by the size of the study area [31]. Thus, the existing related methods have their shortcomings. Some scholars have identified the relevant shortcomings and tried to combine multiple

methods to study the regional spatial conflict situation, such as by analyzing the regional ecosystem service value change under the framework of the "ecology-production-life" theory [32], using the landscape pattern index to quickly and accurately identify the regional "ecology-production-life" spatial evolution [33], or studying the response of ecosystem service function to the landscape pattern change caused by land use transformation [34]. However, there is no study combining these three theories, leading to one-sided results in the identification of ecological-economic space conflicts.

Most of the existing studies take cities and counties as the research units or take the whole country as the subject investigated. Few studies have researched the spatial conflict situation in metropolitan areas, which is a novel and advanced form of territorial spatial organization [4,33,34]; the role of the metropolitan is becoming apparent in regional economic development. Along with the rapid development of metropolitan areas, there are frequent transfers between various land types [35,36]. The original spatial pattern breaks, leading to many spatial conflict problems. Metropolitan areas have become an important driving force behind China's economic growth and contribute to the coordinated development of regional space, which is increasingly receiving national attention [37].

Located in the Yangtze River Delta, the Nanjing metropolitan area is China's first inter-provincial metropolitan area. In February 2021, the Nanjing metropolitan area became the first metropolitan area plan in China to be officially approved by the National Development and Reform Commission, marking a further increase in the strategic status of the Nanjing metropolitan area. Owing to its superior natural geographical conditions, the Nanjing metropolitan area economy developed rapidly. At the same time, conflicts caused by environmental protection and economic development are increasingly intensified. Therefore, there is an urgent need to study the actual situation and the relevant influencing factors of ecological-economic space conflicts [38]. To explore the conflict caused by the disharmony and inconsistency between ecological and economic space, this study used the comprehensive evaluation model of ecological-economic space conflict to identify the severity using the random forest method. This study provides an important realistic basis for the high-quality development of the Nanjing metropolitan area and new ideas for studying the sustainable development path of metropolitan areas in China.

## 2. Materials and Methods

### 2.1. Study Area

Situated at the lower reaches of the Yangtze River (Figure 1), the Nanjing metropolitan area has a warm and humid climate and is blessed with abundant natural resources. The southern part of the metropolitan area has a high forest cover and rich forest resources. The northern and central parts are mainly plains, which are suitable for agricultural development. The metropolitan area has well-developed river systems with two important drainage systems, the Yangtze River and the Huai River, and large lakes such as Hung-tse Lake and Gaoyou Lake. Now, the Nanjing metropolitan area has 8 cities and 2 districts, including Nanjing, Zhenjiang, Yangzhou, Huaian, Maanshan, Chuzhou, Wuhu, Xuancheng, Liyang, and Jintan in Changzhou. The total area of the Nanjing metropolitan is over 65,000 square km and the population is more than 35 million. Its GDP is above 4.6 trillion yuan, making the Nanjing metropolitan area one of the most important economic centers in the Yangtze River Delta. The rapid development of the social economy in the Nanjing metropolitan area has brought great pressure and challenges to the local environment, making the ecological-economic space conflict increasingly fierce.

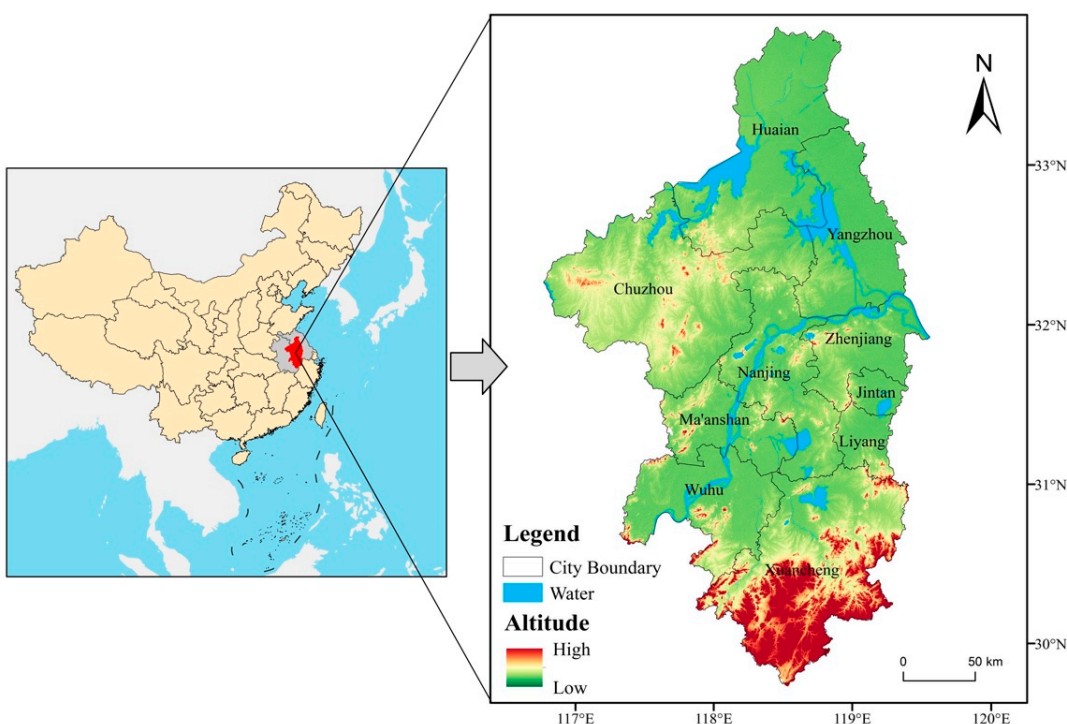

**Figure 1.** Administrative division and terrain of the study area.

*2.2. Research Framework*

　　The essence of spatial conflict is the game of spatial resource possession between the conflicting parties. Along with the development and use of spatial resources, the original spatial pattern will also change, leading to changes in spatial functions and thus changes in the spatial carrying capacity of the region. On this basis, we divide the spatial use mode into three different use types: spatial resource use, spatial function use, and spatial capacity use and construct a spatial conflict identification system by combining relevant previous studies [39]. Then, we classified spatial conflicts into five levels to identify and analyze the ecological-economic conflicts in the Nanjing metropolitan area. Finally, the study used the random forest method to measure the contribution of the relevant driving factors to analyze their importance. Our study research framework is shown in Figure 2.

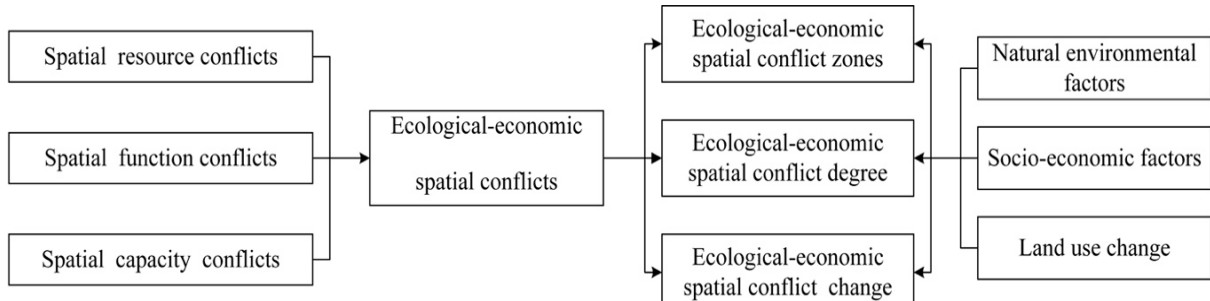

**Figure 2.** Research framework diagram.

*2.3. Data Collection*

　　Table 1 shows the source of the data; our data can be divided into three main categories. First, the study divided the land use data into six types according to LUCC standards [40]; the specific situation of land use is shown in Figure 3. Second, the natural environment data are mainly from relevant satellite remote-sensing image data. Third, the spatial resolution of

the socioeconomic data is 1 km and was mostly provided by the Resource and Environment Data Sharing Center of the Chinese Academy of Sciences.

**Table 1.** Data sources.

| Data | Resolution Data | Available Time Interval | Source |
|---|---|---|---|
| Land Use Data | 1 km × 1 km | 2010, 2015, 2020 | Resource and Environmental Science and Data Center (https://www.resdc.cn (accessed on 25 January 2020)) |
| Net primary productivity (NPP) | 0.5 km × 0.5 km | 2010–2020 | Product of MOD17A3H estimated by moderate resolution imaging spectroradiometer (MODIS) images (http://www.noaa.gov/ (accessed on 26 January 2020)) |
| Normalized difference vegetation index (Ndvi) | 1 km × 1 km | 2010–2020 | MYDND1M China 500M (http://www.noaa.gov/ (accessed on 11 January 2021)) |
| Fine particulate matter (PM2.5) | 1 km × 1 km | 2010–2020 | https://doi.org/10.5281/zenodo.6372847 (accessed on 18 March 2022) |
| Nighttime light (NtL) | 1 km × 1 km | 2010–2020 | NOAA (https://ngdc.noaa.gov/eog/dmsp/downloadV4 composites.html (accessed on 7 April 2022)) |
| Gross domestic product (GDP) | 1 km × 1 km | 2010, 2015, 2019 | Resource and Environmental Science and Data Center (https://www.resdc.cn (accessed on 17 April 2022)) |
| Population data (Pop) | 1 km × 1 km | 2010–2020 | Worldpop (https://www.worldpop.org/ (accessed on 27 March 2021)) |

Since the Chinese Academy of Sciences has not yet given the spatial data of China's GDP in 2020, we choose to use the spatial data of China's GDP in 2019 instead.

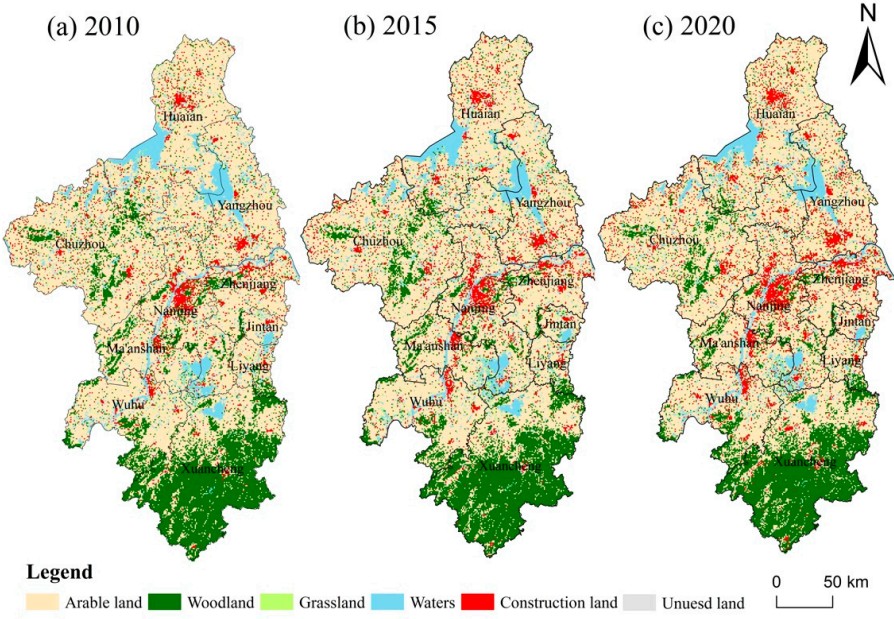

**Figure 3.** Land use map of Nanjing metropolitan area. (**a**) 2010 land use map; (**b**) 2015 land use map; (**c**) 2020 land use map.

### 2.4. Indicator Construction

In this study, spatial use was divided into three types, corresponding to the spatial resource development process, the spatial function change situation, and the spatial carrying capacity change situation. We then selected the corresponding evaluation indexes from the natural factors and socio-economic factors to construct an ecological-economic spatial conflict evaluation index system. We selected relevant data from 2010, 2015, and 2020 and conducted standardized processing before the calculation to prevent the possible uncertain impact of different data values on the overall operation. The composition of the indicator system is shown in Figure 4. The entropy weighting method was used to calculate the weights of resource conflict (RC), function conflict (FC), and capacity conflict (CC) to derive the spatial conflict value for that year [39]. All of these are shown in Table 2. The

entropy weight method is an objective empowerment way to calculate the weight through the information entropy, which is through the dispersion degree of the original data of each index. It can effectively avoid the deviation caused by subjective factors and improve the credibility and accuracy of the index weight value [41,42]. We used the weighted average of the 3 option weights as the final weight of the study, with resource conflict (RC), function conflict (FC), and capacity conflict (CC) having final weight values of 0.32, 0.42, and 0.25. Finally, the study used ArcGIS 10.7 and applied equal interval classification to classify the Nanjing metropolitan area 2010–2020 ecosystem service demand index into five classes: highest-conflict, high-conflict, medium-conflict, low-conflict, and lowest-conflict.

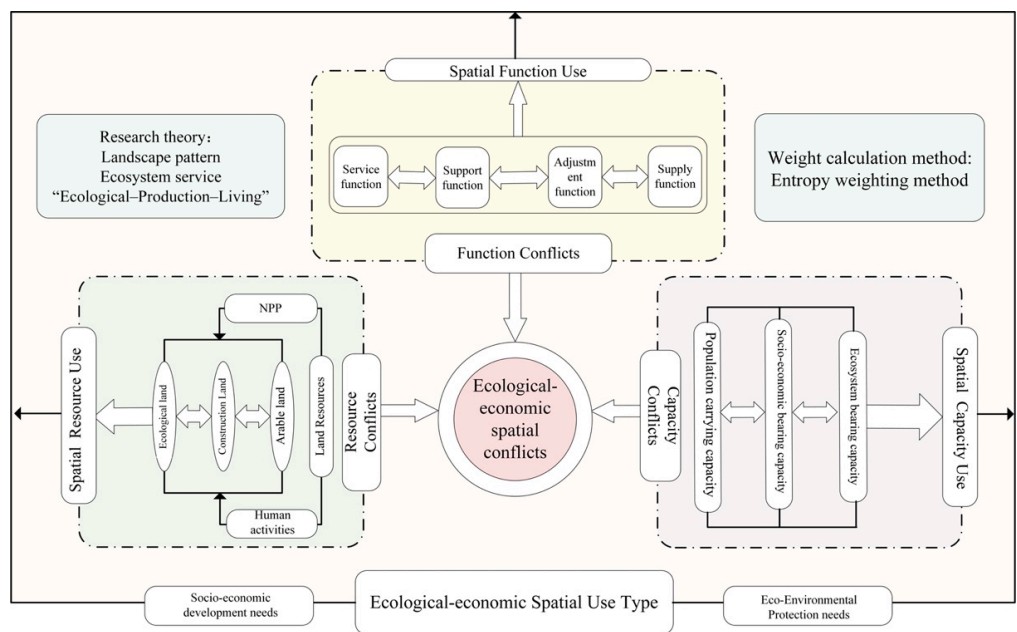

**Figure 4.** Indicator system diagram.

**Table 2.** Ecological-economic spatial conflict evaluation Indicator system.

| Indicator Type | Standard Layer | Indicator Layer | Indicator Attribute | Indicator Description |
|---|---|---|---|---|
| Resource Conflict (RC) | Land use conflict (RC1) | Landscape aggregation index (AI) | Negative | Reflects the conflict between cultivated land resources and construction land |
| | | Landscape sprawl index (Contag) | Negative | Reflects the conflict between ecological land resources and construction land |
| | Human activities clash with natural resources (RC2) | NPP data | Negative | Reflects the vegetation regeneration capacity |
| | | Construction–land density reaction | positive | Reaction to the consumption of land resources |
| Function Conflict (FC) | Supply and demand of ecosystem services conflict (FC1) | Supply of ecosystem services | Negative | Reflects the supply and demand of ecosystem services |
| | | Demand of ecosystem services | positive | |
| | Carbon-fixing capacity conflicts with carbon emissions (FC2) | Carbon emissions | positive | Reflect carbon emissions and carbon storage |
| | | Carbon sequestration | Negative | |
| Capacity Conflict (CC) | Biodiversity conflict (CC1) | Habitat quality | Negative | Reflects species richness through biodiversity |
| | Economic and environmental conflict (CC2) | GDP | positive | Reflects the economic development situation |
| | | PM2.5 | positive | Reflects the air environmental quality situation |
| | | Pop | positive | Reflects the size of the population |
| | | Ndvi data | Negative | Reflects the vegetation coverage situation |

*2.5. Interpretation and Calculation of Indicators*

Resource conflict (RC) is mainly expressed as the conflicts between human-led economic activities, food production, and other processes on other biological and resource supplies. Among them, RC1 uses AI to measure the conflicts between arable land and construction land, the degree of cultivated land fragmentation to reflect the conflict between construction land and arable land, Contag to measure the degree of ecological land and construction land, and the degree of fragmentation to respond to the conflicts between construction land and ecological land. The degree of ecological land fragmentation responds to the spatial connectivity of ecological land by the encroachment of construction land [43,44]. RC2 uses NPP data and the density of built-up land to reflect the influence of urban development on vegetation regeneration capacity. The space occupied by human socio-economic activities undoubtedly affects the growth of the original ecological vegetation. The NPP data were used to reflect the spatial situation of vegetation growth activities on the surface, whereas the density of built-up land reflects the distribution intensity of human economic activities on the surface [45,46].

$$AI = \left[ \sum_{i=1}^{m} \left( \frac{g_{ii}}{\max g_{ii}} \right) p_i \right] \times 100 \tag{1}$$

where $g$ is the number of nodes between image elements of patch type $i$ based on the single-fold method and $\max g_{ii}$ is the maximum number of nodes between image elements of patch type $i$ based on the single-fold method. $P_i$ is the area proportion of patch type $i$ in landscape.

$$CONTAG = \left\{ 1 + \frac{\sum\limits_{i=1}^{m} \sum\limits_{k=1}^{m} \left[ p_i \left( g_{ik} / \sum\limits_{k=1}^{m} g_{ik} \right) \right] \times \left[ \ln p_i \left( g_{ik} / \sum\limits_{k=1}^{m} g_{ik} \right) \right]}{2 \ln(m)} \right\} \times 100 \tag{2}$$

where $P_i$ is the proportion of area of patch type $i$ in the landscape, $g_{ik}$ is the number of nodes between patch type $i$ and patch type $k$ on the basis of the doubling method, and m is the number of patch types in the landscape, including those in the landscape boundary.

Functional conflict (FC) is mainly reflected in the overlapping area of the economic activity area and ecological function area. The overlap results in conflict caused by social and economic production to ecological function disturbance. According to a previous study, we used ecosystem service supply and demand to judge the functional conflict of ecological-economic space [47]. The ecosystem service supply was calculated by the value equivalent method, while the demand for ecosystem service was calculated according to previous studies [25]. Based on previous studies, this study uses nighttime light (NtL) to estimate carbon emissions [48]. Then we used the invest model carbon storage module to calculate carbon reserves. The invest model carbon storage module specific formula is

$$C_{x,t} = \sum_{j=1}^{J} A_{xj} \left( C_{aj} + C_{bj} + C_{sj} + C_{dj} \right) \tag{3}$$

$C_x$ is the carbon stock of region x in t, $A_{xj}$ is the area of land cover type $j$ in region $x$, and $C_{aj}$, $C_{bj}$, $C_{sj}$, and $C_{dj}$ represent the above-ground carbon density, below-ground carbon density, soil carbon density, and dead organic matter carbon density of land cover type $j$, respectively [49].

Capacity conflict (CC) mainly occurs in the process of space evolution between human activities and the ecological environment. CC1 uses the invest model of the habitat quality module to calculate the impact of human activities on the bearing capacity of the natural environment [50]. Based on previous studies, CC2 uses GDP, PM2.5, Pop, and Ndvi to reflect the conflict between environmental protection and economic development [39].

GDP and Pop represent the capacity of economic development while the Ndvi and PM2.5 represent the capacity of ecological protection.

The habitat quality module formula is

$$Q_{xj} = H_j \left( \frac{K^z}{\left( D_{xj}^z + K^z \right)} \right) \tag{4}$$

where $Q_{xj}$ is the habitat quality index of raster cell $x$ in land use type $j$ and $H_j$ is the habitat suitability of land use type $j$. The value range is [0, 1]. The closer the value is to 1, the higher the habitat quality. $D_{xj}$ is the degradation degree of raster cell $x$ in land use/cover type $j$. $K$ is the half-saturation constant, which is usually half of the maximum degradation degree; the default value is 0.5. $z$ is the normalization constant, which is the default parameter of the model, and takes the model definition value of 2.5 [51].

### 2.6. Data Sources and Methods of Driving Factor

Eight potential driving factors were selected for analysis as potential causes of ecological-economic spatial conflicts that may affect the Nanjing metropolitan area [52]. These driving factors include two main aspects: (1) For natural environmental factors, the study selected DEM, distance to water, average annual temperature, and soil as driving factors in the natural environment. DEM, distance to water, and soil type data were obtained from the Chinese Academy of Sciences Resource Environment Data Sharing Center (https://www.resdc.cn/(accessed on 11 March 2022)). The annual average temperature data were obtained from the site data interpolation of China Meteorological Network (http://www.cma.gov.cn/ (accessed on 11 January 2022)). (2) For socio-economic factors such as distance to the major highways, distance to the Nanjing, industrial density (nuclear density of industrial parks and development zones), and distance to the railroad, the data were crawled from Amap.

The random forest algorithm has excellent performance for establishing the nonlinear relationship between input variables and output variables [53]. Based on the principle of random forest, Liang proposed a patch-generated land-use simulation model (PLUS model), which has been successfully applied to dynamic simulation and prediction of land-use change and can analyze the contribution of related drivers to land use change [54]. The role of the LEAS model is to transform the mining of transition rules of each land use type in the PLUS model into a binary classification problem. This is specifically done to calculate the relationship between the growth of each land use type and the associated drivers based on the random forest algorithm and finally output the growth probability $P_{i,k}^d$ of land use type $k$ at cell $i$. The random forest algorithm formula is

$$P_{i,k}^d(x) = \frac{\sum\limits_{n=1}^{M} I(h_n(x) = d)}{M} \tag{5}$$

The value of $d$ is either 0 or 1; when the value of $d$ was 1, there were other land use types changed to land use type $k$, while 0 represents other transitions; $x$ is a vector that consists of multiple driving factors; $I(\cdot)$ is the indicative function of the decision tree set; $h_n(x)$ is the prediction type of the $n$-th decision tree for vector $x$; and M is the total count of decision trees.

Jiang used similar principles to analyze the impact of related factors on potential pollution sites in the Yangtze River Delta [55]. On this basis, we performed spatial superpositions on the ecological-economic spatial conflict zone data in different periods, subtracted the previous spatial conflict zone data from the spatial conflict zone data in the latter period, and extracted the changes to represent the change areas of each spatial conflict level. Then, we used the LEAS module of the PLUS model to mine the influence of each driving factor on the change of each level of conflict area based on the random forest algorithm to evaluate the relative importance of each driver on different types of spatial conflict area and provide sta-

ble and accurate classification results. For a detailed description of the PLUS model, please refer to https://github.com/HPSCIL/Patch-generating_Land_Use_Simulation_Model (accessed on 27 January 2022).

## 3. Results

### 3.1. Spatial Distribution Characteristics of Ecological-Economic Space Conflict

This study used the constructed index system to obtain the ecological-economic spatial conflict distribution characteristics of the Nanjing metropolitan area for the three periods of 2010, 2015, and 2020. Figure 5 and Table 3 show that the ecological-economic spatial conflict in the Nanjing metropolitan area changes substantially from 2010 to 2020. The spatial conflict state in the Nanjing metropolitan area was dominated by low conflict. The proportion of low-conflict areas increased from 58.94% in 2010 to 66.15% in 2020. Meanwhile, the proportion of highest-conflict and high-conflict areas were low. These situations together show that the degree of ecological-economic space conflict in the Nanjing metropolitan area is not serious. However, we found some features of its space-time evolution: First, numerous medium-conflict areas were spread along the edges of the city in 2010. Medium-conflict areas accounted for 24.49% of the period, which gradually stabilized at around 18% over the next decade. Second, the high-conflict and highest-conflict areas were distributed in the city and its surrounding areas, and two larger core conflict areas formed in the central region along the river and northern region; the area proportion of these two conflicts is increasing. Finally, the lowest-conflict areas were mainly in the mountainous zone of Xuancheng and the hilly areas of Chuzhou, but the proportion of low-conflict areas is shrinking.

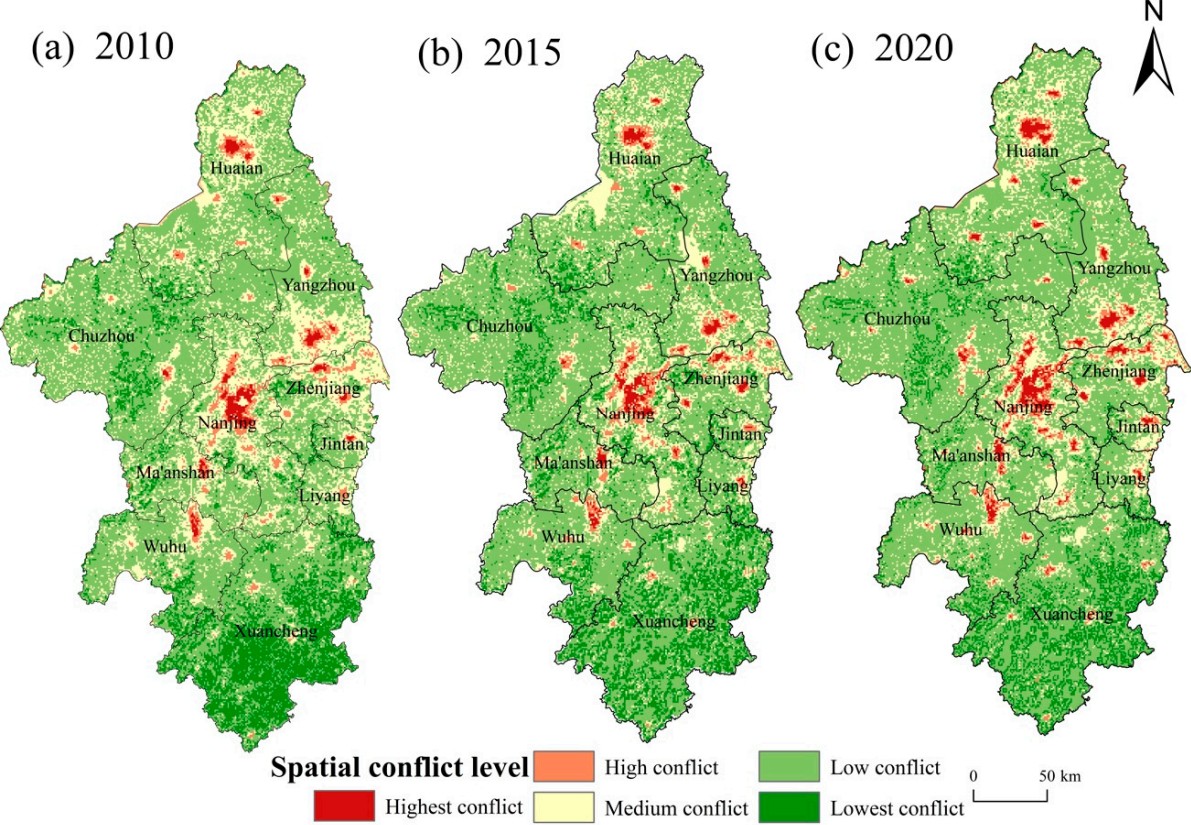

**Figure 5.** Distribution of ecological-economic spatial conflict in Nanjing metropolitan area. (**a**) 2010 spatial conflict map; (**b**) 2015 spatial conflict map; (**c**) 2020 spatial conflict map.

**Table 3.** Area composition of different conflict types.

| Conflict Types | Highest Conflict | High Conflict | Medium Conflict | Low Conflict | Lowest Conflict |
|---|---|---|---|---|---|
| 2010 | 1.07% | 2.95% | 24.49% | 58.94% | 12.56% |
| 2015 | 1.24% | 3.33% | 18.22% | 64% | 13.21% |
| 2020 | 1.94% | 3.69% | 18.98% | 66.15% | 9.24% |

*3.2. Spatial-Temporal Evolution of Ecological-Economic Spatial Conflict*

To investigate the spatial and temporal changes of ecological-economic spatial conflicts in the Nanjing metropolitan area, we created a map of ecological-economic spatial conflict change zone transfer in the Nanjing metropolitan area (Figure 6) between 2010 and 2020. Figure 6 shows that transfers between spaces of different conflict levels were more frequent during 2010–2015, whereas the frequency of transfers decreased remarkably during 2015–2020. Specifically, highest-conflict and high-conflict areas had the least probability of spatial shifts. Shifts between medium-conflict areas, low-conflict areas, and lowest-conflict areas were frequent, especially between medium-conflict areas and low-conflict areas.

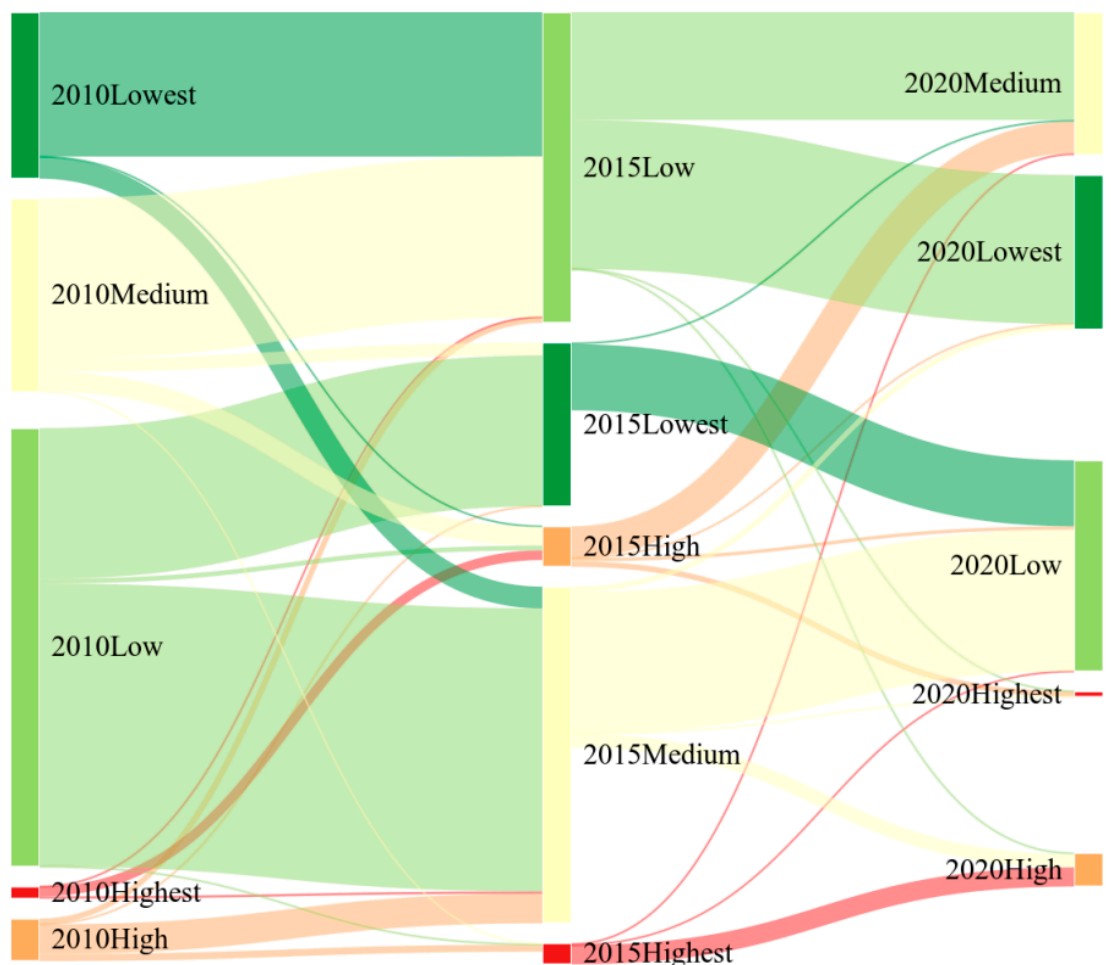

**Figure 6.** Transfer of ecological-economic space conflict change zone in Nanjing metropolitan area from 2010 to 2020 (the color of the highest, high, medium, low, and lowest five different spatial conflict levels is consistent with those in Figure 5).

Hotspot analysis (Figure 7) can fully show the clustering of ecological-economic spatial conflicts in the Nanjing metropolitan area: the deeper the red color is, the higher the degree of clustering in areas with high-conflict values, and vice versa [56]. Figure 7 shows that the spatial agglomeration characteristic of cold and hot spots in the Nanjing metropolitan area

is remarkable. The cold spots in the Nanjing metropolitan area are mostly concentrated in the hilly areas of Chuzhou and the mountainous region of Xuancheng. The overall change is large, with an obvious decrease in cold spots over the last 10 years. The proportion of cold points in 2020 decreased by 22% compared with 2010. The hot spot areas were mainly concentrated around the central part of the metropolitan area along the Yangtze river and the urban area of Huai'an in the north, showing a trend of concentration to the city and its surrounding areas, and the hot spot value has significantly increased.

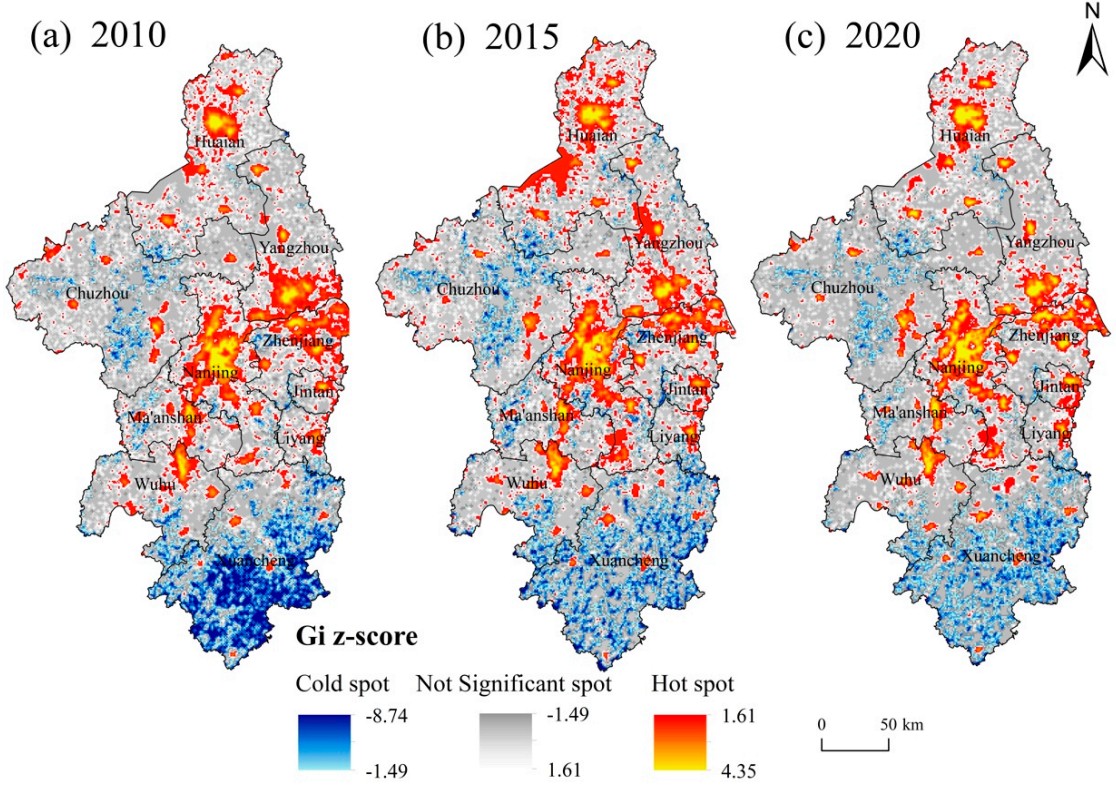

**Figure 7.** Cold−hot spot map of ecological-economic space conflict for 2010−2020. (**a**) 2010 cold−hot spot map; (**b**) 2015 cold−hot spot map; (**c**) 2020 cold−hot spot map.

### 3.3. Factors Influencing the Change of Ecological-Economic Spatial Conflict in Nanjing Metropolitan Area

3.3.1. Effect of Land Use on Ecological-Economic Space Conflict

Based on the analysis of the proportion of land use type in different conflict levels (Figure 8), construction land is the main land use type in the highest-conflict zone and high-conflict zone in Nanjing metropolitan area from 2010 to 2020. Construction land in 2010 only accounted for 60% of the high-conflict area, while this proportion rose to 79% by 2020. In the highest-conflict areas, the proportion of construction land has been above 90%, and reached 98% in 2020. The medium-conflict areas are mostly located around various cities and mainly show spatial conflict between arable land and construction land; that is, the proportion of arable land and construction land is the highest. As an important carrier of many socio-economic activities, the coastal parts of lake areas are disturbed by human activities and their ecological function is weakened, which makes them prone to medium spatial conflicts. As arable land is the biggest land type in the Nanjing metropolitan area, arable land is also the main type of land in the low-conflict zones, where its share is consistently around 70%. According to the current situation, the areas with the lowest conflict are mainly ecological lands such as forest and grassland, with the highest percentage of woodlands consistently being around 70% (Table 4).

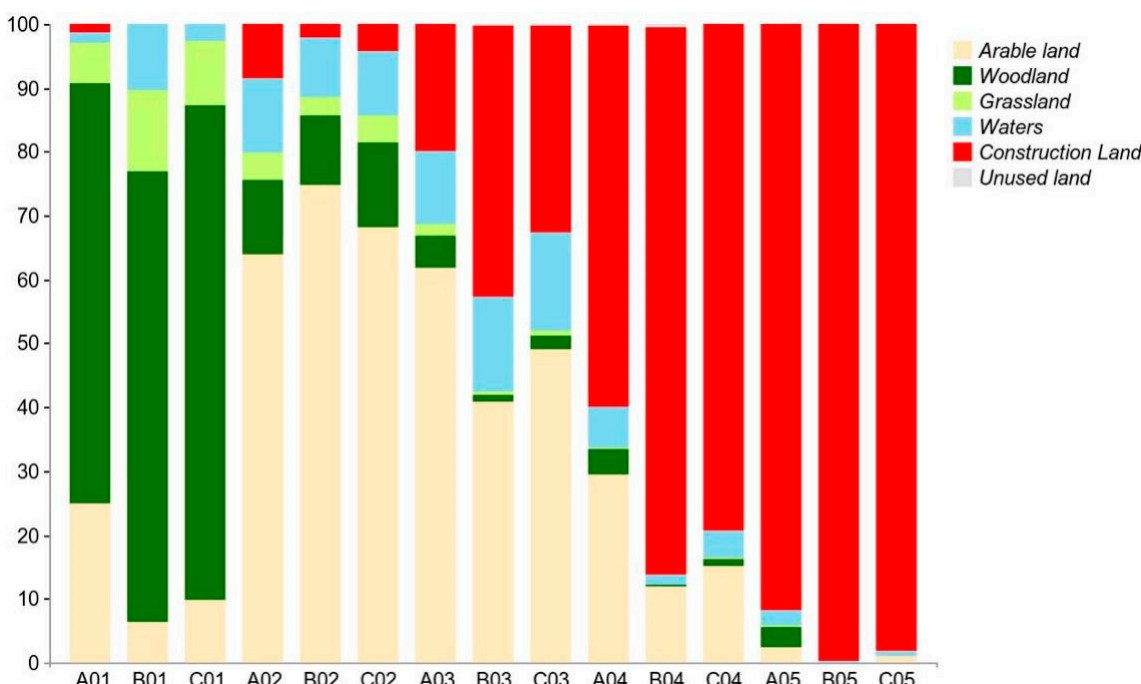

**Figure 8.** Composition of land use in different years of each conflict types (A, B, and C refer to 2010, 2015, and 2020; 01, 02, 03, 04, 05 refer to lowest, low, medium, high, highest spatial conflict levels).

**Table 4.** The proportion of different land use types in each conflict level.

| Time | Land Use Type | Lowest Conflict | Low Conflict | Medium Conflict | High Conflict | Highest Conflict |
|------|---------------|-----------------|--------------|-----------------|---------------|------------------|
| 2010 | Arable land | 24.97 | 63.97 | 61.76 | 29.31 | 2.34 |
| | Woodland | 65.73 | 11.68 | 5.11 | 3.97 | 3.21 |
| | Grassland | 6.46 | 4.26 | 1.83 | 0.26 | 0.15 |
| | Waters | 1.57 | 11.57 | 11.52 | 6.52 | 2.63 |
| | Construction Land | 1.20 | 8.44 | 19.68 | 59.57 | 91.68 |
| | Unused land | 0.07 | 0.08 | 0.10 | 0.37 | 0 |
| 2015 | Arable land | 6.3 | 74.71 | 40.90 | 11.80 | 0 |
| | Woodland | 70.51 | 10.86 | 1.12 | 0.33 | 0 |
| | Grassland | 12.89 | 3.08 | 0.53 | 0.09 | 0 |
| | Waters | 10.20 | 9.23 | 14.79 | 1.42 | 0.25 |
| | Construction Land | 0.06 | 2.09 | 42.36 | 85.88 | 99.75 |
| | Unused land | 0.04 | 0.03 | 0.29 | 0.47 | 0 |
| 2020 | Arable land | 9.67 | 68.17 | 48.93 | 15.12 | 0.98 |
| | Woodland | 77.54 | 13.20 | 2.28 | 1.07 | 0 |
| | Grassland | 10.22 | 4.18 | 0.77 | 0.17 | 0 |
| | Waters | 2.45 | 10.07 | 15.46 | 4.33 | 0.98 |
| | Construction Land | 0.10 | 4.31 | 32.35 | 79.22 | 98.04 |
| | Unused land | 0.02 | 0.06 | 0.21 | 0.09 | 0 |

### 3.3.2. Driving Factor of Changes in the Ecological-Economic Space Conflict

This study was based on previous studies that used similar principles to the LEAS model, which is based on the random forest method to rank the ecological-economic spatial conflict driving factors in the Nanjing metropolitan area. Then, we analyzed the contribution of different driving factors to different conflict rank regions. The number of decision trees was set as 50, the sampling rate set as 0.1, and the total sample size was 8939; also, this study used RMSE to reflect the accuracy of random forests. In general,

the precision values of the random forest were inversely proportional to the RMSE values (Table 5).

**Table 5.** Random forest accuracy by RMSE.

| Time | Highest Conflict | High Conflict | Medium Conflict | Low Conflict | Lowest Conflict |
|---|---|---|---|---|---|
| 2010–2015 | 0.03 | 0.07 | 0.14 | 0.17 | 0.12 |
| 2015–2020 | 0.07 | 0.15 | 0.17 | 0.13 | 0.09 |
| 2010–2020 | 0.07 | 0.14 | 0.16 | 0.13 | 0.09 |

Just as shown in Figure 9 and Table 6, the different influencing factors affect each conflict state to different degrees, and the influence of DEM on each conflict-level area is substantial, especially for the lowest-conflict areas. In the 2010–2015 period, the contribution weight of DEM to the lowest-conflict region and low-conflict region both exceeded 0.2. In the 2015–2020 period, the contribution weight of DEM to the lowest-conflict region reached 0.39. As a whole, the contribution weights of DEM to the lowest conflict region and low conflict region during 2010–2020 were 0.18 and 0.24, both of which are at the top of the contribution scale. The distance to water is also an important driving factor for lowest-conflict and low-conflict areas. Overall, industrial density had the greatest impact on the expansion of high and highest conflict areas in the 2010–2020 period, with contribution weights of 0.2 and 0.23, which were above the other driving factors. However, between 2010 and 2015, the largest contributor to the expansion of highest-conflict areas was distance to railroads with a weight of 0.22, slightly higher than the industrial density (0.18), while the distance to highways contributed much less than the distance to railroads. The weight of the contribution of industrial density to the highest-conflict areas was 0.23 for the 2015–2020 period, which was at the top of each driving factor. The degree of contribution of soil type, the distance to the Nanjing, and the annual average temperature were also not remarkable, indicating that the development of the Nanjing metropolitan area relies more on the remaining characteristics.

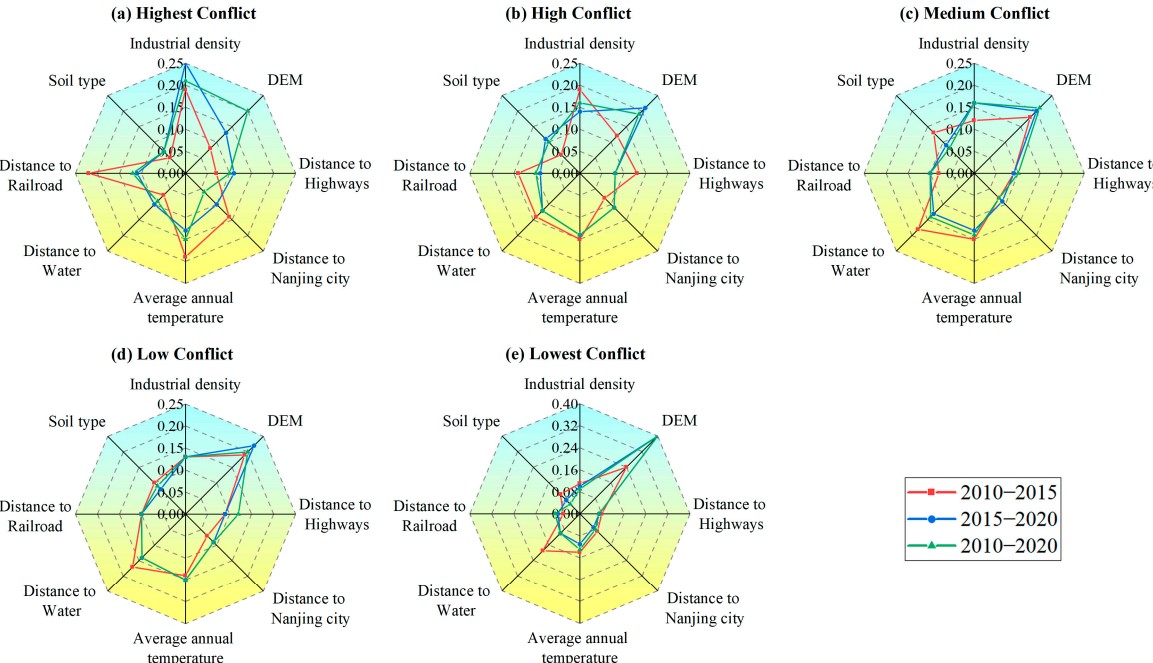

**Figure 9.** Driving factors of the ecological-economic spatial change.

**Table 6.** Contribution weight values of related driving factors.

| Time | Factors | Highest Conflict | High Conflict | Medium Conflict | Low Conflict | Lowest Conflict |
|---|---|---|---|---|---|---|
| 2010–2015 | Industrial density | 0.18 | 0.18 | 0.12 | 0.13 | 0.12 |
| | DEM | 0.09 | 0.13 | 0.19 | 0.2 | 0.24 |
| | Distance to highways | 0.06 | 0.13 | 0.09 | 0.1 | 0.09 |
| | Distance to Nanjing city | 0.13 | 0.08 | 0.07 | 0.07 | 0.09 |
| | Average annual temperature | 0.19 | 0.17 | 0.14 | 0.13 | 0.13 |
| | Distance to water | 0.07 | 0.13 | 0.18 | 0.16 | 0.19 |
| | Distance to railroad | 0.22 | 0.12 | 0.08 | 0.11 | 0.05 |
| | Soil type | 0.05 | 0.07 | 0.12 | 0.1 | 0.09 |
| 2015–2020 | Industrial density | 0.23 | 0.15 | 0.16 | 0.11 | 0.11 |
| | DEM | 0.15 | 0.19 | 0.19 | 0.21 | 0.39 |
| | Distance to highways | 0.11 | 0.1 | 0.09 | 0.12 | 0.06 |
| | Distance to Nanjing city | 0.07 | 0.1 | 0.1 | 0.1 | 0.09 |
| | Average annual temperature | 0.13 | 0.14 | 0.13 | 0.15 | 0.11 |
| | Distance to water | 0.1 | 0.13 | 0.12 | 0.14 | 0.1 |
| | Distance to railroad | 0.14 | 0.1 | 0.1 | 0.1 | 0.08 |
| | Soil type | 0.07 | 0.09 | 0.1 | 0.08 | 0.05 |
| 2010–2020 | Industrial density | 0.2 | 0.23 | 0.13 | 0.15 | 0.13 |
| | DEM | 0.12 | 0.1 | 0.22 | 0.18 | 0.24 |
| | Distance to highways | 0.09 | 0.09 | 0.08 | 0.1 | 0.08 |
| | Distance to Nanjing city | 0.09 | 0.08 | 0.08 | 0.08 | 0.08 |
| | Average annual temperature | 0.17 | 0.17 | 0.16 | 0.14 | 0.17 |
| | Distance to water | 0.12 | 0.11 | 0.12 | 0.15 | 0.11 |
| | Distance to railroad | 0.15 | 0.13 | 0.1 | 0.11 | 0.09 |
| | Soil type | 0.07 | 0.09 | 0.11 | 0.09 | 0.1 |

## 4. Discussion

In the context of ecological civilization construction, the identification of ecological-economic spatial conflict and research on influencing factors have become an important prerequisite for high-quality development, which may help policymakers and stakeholders to conduct investigation and adjustment work. In this study, all the data were derived from remote sensing satellite data and their deductive data to ensure accuracy and authenticity. Researchers have carried out many related studies on space conflict identification, which provided us with the relevant theoretical basis and methods for the identification of ecological-economic space conflicts. However, due to the defects of relevant theories, there are still some deficiencies in the identification of ecological-economic space conflicts. Based on previous studies, we tried to create a comprehensive, integrated index system to reflect the ecological-economic space conflict in the Nanjing metropolitan area as truthfully and thoroughly as possible. Furthermore, we analyze the extent of different drivers of the ecological-economic spatial conflict in the Nanjing metropolitan area through the latest random forest methods [55].

### 4.1. The overall Pattern of Ecological-Economic Spatial Conflicts Is Stabilizing, but the Lowest Conflict Areas Will Gradually Shrink

Our study shows that the situation of ecological-economic spatial conflict in the Nanjing metropolitan area has changed significantly over the past decade. This is reflected in the continuous increase of the highest-conflict and high-conflict zones, while the area of the lowest-conflict zone continues to decline. The percentage of area in the lowest conflict region in 2020 was only 9.24%. The area proportion of low-conflict zones increased from 58.94% in 2010 to 66.15%, and the area share of medium-conflict zones gradually decreased from 24.49% in 2010 before stabilizing at about 18% in 2020. In terms of the transfer of spatial conflict regions, the frequency of transfer is decreasing, indicating that the overall spatial conflict patterns gradually stabilize. Among them, the probability of

the highest-conflict and high-conflict areas undergoing spatial transfer is minimal, and the highest-conflict and high-conflict areas will be difficult to change once formed. Medium-conflict and low-conflict areas are the most likely to be affected and change. In particular, the transfer of medium-conflict zones from 2010 to 2015 was very frequent. The spatial structure of medium-conflict zones and low-conflict zones was still not stable enough, the spatial structure of these areas was not stable enough, and changes in relevant policies or new development projects can change the local spatial conflict situation. From the change on the cold–hot spot maps, we can further see that the hot spots eventually contracted to be near the city, especially around the central part of the metropolitan area along the Yangtze river and the urban area of Huai'an in the north. This further proves that the ecological-economic space conflict situation in the Nanjing metropolitan area is effectively controlled and that the high-conflict level areas are restricted to the city zone. At the same time, we also note that with the development of the economy, the proportion of the cold spot areas will continue to decrease; in particular, the Xuancheng area decreased at a significant rate, which means that the pressure on ecological and environmental protection will continue to increase.

*4.2. Different Land Use Types Have Different Effects on Changes in Ecological-Economic Spatial Conflicts*

The spatial overlap of different functional requirements leads to spatial conflict. As the most common and direct influencing factor of spatial conflicts, land use patterns have always had a profound effect on the evolution of spatial conflicts [57]. Based on the proportion of different land use types at different conflict levels, construction land is undoubtedly the most important trigger of ecological-economic conflicts, because it must carry out most socio-economic activities, which further weakens the ecological functions of this land use type. The medium-conflict and low-conflict areas are mainly made up of arable land and water areas. Arable and water areas are mostly on the edge area of human activities. By rational planning of the use of these two types of land, industrial development can be achieved without damaging the ecosystem. However, they are inherently less stable and vulnerable to the influence of surrounding areas, so strict spatial boundary planning control of these land use types is needed to reduce the impact of adjacent units on them [12]. Pure ecological land types such as woodland and grassland are the main land types in the lowest-conflict areas, and strict protection measures should be taken so that ecological land can rebuild the regional ecological security pattern. Developing ecotourism industries and other industries that have less effect on the ecological environment while improving land use efficiency as much as possible is also important [58].

*4.3. The Development of Eco-Economic Spatial Conflict Mitigation Measures Needs to Be Tailored to Local Conditions*

According to the analysis of the relevant factors, we found that the relevant driving factors of spatial conflicts showed multi-level features, meaning that the development of conflict reconciliation countermeasures should also be adjusted to local conditions [59]. The distribution of industries largely influences the proliferation of both high-conflict and highest-conflict areas, so the study can conclude that the construction of industrial parks will quickly change the local conflict state and will lead to the rapid enhancement of socio-economic functions as well as the decline of ecological functions [60]. Hence, industrial parks must be built in and around cities as much as possible, away from ecological reserves or ecologically fragile areas. This was also confirmed in a study by Wang et al. [61]. DEM is the factor that contributes most to the lowest-conflict and low-conflict areas. Given that the Nanjing metropolitan high-altitude areas are mountainous forests, they are not amenable to socio-economic activities. Moreover, most of these areas belong to ecological protection zones, so the distribution of altitude highly overlaps with the distribution of lowest-conflict and low-conflict areas. At the same time, the low-elevation Nanjing metropolitan areas are mostly plains, which are suitable for socio-economic activities and agricultural production. Therefore, the spatial utilization of the Nanjing metropolitan area should be developed

rationally according to local conditions. In addition, relevant policies or major constructions can have a remarkable effect on the conflict situation in that year. For example, railroads contributed the most to the expansion of high-conflict areas during 2010–2015. High-speed railway stations were mostly established near suburbs in the past and the establishment of high-speed railway stations could drive the rapid development of the surrounding area, which made the ecological-economic spatial conflicts near high-speed railway stations intensify rapidly.

### 4.4. Findings and Policy Suggestions

In sum, we found that due to better ecological background conditions, the degree of ecological-economic spatial conflict in the Nanjing metropolitan area is not serious and mainly dominated by low conflict. However, the Nanjing metropolitan area's ecological environment pressure will gradually increase with the continuous development of the social economy, so relevant ecological environment protection measures, such as set natural protection and water protection, should be implemented to ensure that the low-conflict areas no longer drop. Along with the continuous development of the city, the high-conflict and highest-conflict areas will inevitably increase. Based on the analysis of the changes in spatial conflict over the past 10 years and related influencing factors, we find that the most advisable way to alleviate the ecological-economic space conflict in the Nanjing metropolitan area is to, first, limit the disorderly growth of the urban area and strictly implement the requirements of ecological civilization construction and land space planning, second, protect the relevant ecological areas, and third, lock the eco-economic space conflict area in the city and its surrounding areas. This can maximize the avoidance of disorderly expansion similar to 2010 to ensure the normal operation of other ecological spaces and achieve the overall ecological-economic harmony state. In so doing, the Nanjing metropolitan circle can finally achieve the goal of sustainable development. Our conclusion is consistent with the findings of Zhang and Xu [62,63], as well as with the current mainstream solution view of spatial conflict. Scholars now believe that since spatial conflict is inevitable, the solution to spatial conflict should shift from traditional confrontation and elimination to guidance and weakening and should give priority to the legitimate needs of human social development [64–66].

### 4.5. Implications and Limitations

To diagnose Nanjing metropolitan area ecological-economic spatial conflicts, identify them, and determine their intensity, several problems remain to be solved. First, the theoretical understanding of spatial conflict must be strengthened. Most existing studies consider the overlap of different functions of space to be a conflict [67,68]. Most existing studies also believe that an important cause of spatial conflicts is the overlap of different spatial functions. However, a space with only a single function is rare in reality. Therefore, subsequent studies need to systematically discuss the forms and connotations of spatial conflicts. Second, although this study attempted to use spatial data to establish a comprehensive system of indicators to measure ecological-economic spatial conflicts to compensate for the shortcomings of existing studies, we have only analyzed spatial conflicts from an ecological-economic perspective. Space also has cultural, aesthetic, and other social functions, and analyzing these social spatial functions requires the reuse of panel data or through methods such as questionnaires. Our future research may explore spatial conflict from the perspective of civil rights protection and social spatial equity [69]. However, the method for considering these functions in the study of spatial conflicts remains to be explored.

## 5. Conclusions

This study identified and diagnosed the intensity of ecological-economic spatial conflicts in the Nanjing metropolitan area in 2010, 2015, and 2020. The degree of contribution of selected natural environmental and socio-economic factors to the spatial conflict changes

was analyzed via the random forest method. The main conclusions of this paper are as follows:

(1) From 2010 to 2020, the ecological-economic space conflict in the Nanjing metropolitan area changed considerably. The spatial conflict status of the Nanjing metropolitan area was dominated by low conflict areas, and the lowest-conflict areas were mainly concentrated in the hilly areas of Chuzhou and the mountainous areas of Xuancheng. High-conflict and highest-conflict areas had the lowest proportion and were mainly concentrated in urban areas, while two large conflict areas formed in the central and northern regions of the metropolitan area. The proportion of medium-conflict areas is larger and mainly concentrated in the urban periphery.

(2) The change in land use has a substantial effect on spatial conflicts. In general, the main land use types in the lowest-conflict zone are forest land and arable land, the main land use types in the low-conflict and medium-conflict zones are arable land, and the high-conflict and highest-conflict zones consist mainly of construction land. Therefore, spatial conflicts are easily triggered or intensified by disorderly urban expansion, whereas the presence of ecological land can mitigate spatial conflicts.

(3) The relevant driving factors of spatial conflicts showed multi-level features. The factor that contributed most to the lowest-conflict and low-conflict areas was DEM, and the factor that contributed most to the highest-conflict and high-conflict areas was industrial density. However, the situation varied from year to year and from region to region, so the development of conflict reconciliation countermeasures needs to be tailored to local conditions.

Finally, this paper offers suggestions to help the Nanjing metropolitan area achieve sustainable development. As a typical metropolitan area in China, the sustainability and spatial conflict situation of the Nanjing metropolitan area needs more attention.

**Author Contributions:** Conceptualization, W.C.; Formal analysis, J.C.; Methodology, J.C. and J.M.; Supervision, Y.X.; Validation, W.C.; Visualization, J.C. and X.F.; Writing—original draft, J.C.; Writing—review and editing, J.C. and D.M. All authors have read and agreed to the published version of the manuscript.

**Funding:** This work was funded by the National Natural Science Foundation of China (41571124).

**Data Availability Statement:** Not applicable.

**Conflicts of Interest:** The authors declare no conflict of interest.

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
