# Peer review of "The Identification and Driving Factor Analysis of Ecological-Economi Spatial Conflict in Nanjing Metropolitan Area Based on Remote Sensing Data"

_remotesensing, doi:10.3390/rs14225864_

Round 1

Reviewer 1 Report

1. It is difficult to understand the concept of “spatial conflict” in the paper. I think some examples could help readers understand “spatial conflict” and why authors study this topic.

2. The motivation of the paper is not clear. The analysis of the defects of the existing studies should be improved. For example, what is “complex mechanism” Eco-economic spatial conflict. Why should we care about the “complex mechanism”? A well-organized paper helps authors express their opinions.

3. What is “spatial system”? It seems like some important terms are just used without any explanation.

4. Picture 2 looks beautiful, but I found noting related to “five levels” and “relevant driving factors”. The picture of the research framework should be logical and easy to understand.

5 “According to the complexity of the spatial system, the study divides the spatial system into three subsystems: Resource subsystems, Function subsystems, and Capacity subsystems.” I think the complexity of a space system does not justify its division into three subsystems.

Author Response

Appended to this letter is our point-by-point response to the comments raised by the reviewers.We would like also to thank you for allowing us to resubmit a revised copy of the manuscript.

Reviewer 2 Report

The title must be changed to become more intelligible.

The methodology is complex, but the results and discussions are not well presented. It isn't easy to extract the results of the study from the text. 

Fig.7 and Fig.8 are not s easy to read, please try a different way to express results.

 Fig. 6 - change the title of the map.... Cold Hot Spot of ecological- economic space conflict....probably it is best to write "Spatial distribution of the ecological-economical conflict? 

Discussions: this chapter is written in a general manner, it should be improved in order to emphasize the original ideas of the study and to be cited by other authors. Also, when you interpret the results it is necessary to say in which area the phenomenon is visible...in the north , south.etc.

The some for the conclusions.

Overall, we see that there is a lot of work here, but the discussions and the presentation of the results must be improved. We want the conclusions to be clearer and original aspects of the study to be easy to spot in the text.

In the end, we consider that there is lot of work that can be published, only after the text gets less general , but more specific, clear, and the results better presented.

Author Response

(The authors gave the same response as above.)

Reviewer 3 Report

The manuscript tries to formulate a comprehensive, integrated index system to reflect the ecological-economic space conflict of different spatial regions and land use in Nanjing, China. It combines with the use of random forest techniques for obtaining relevant spatial maps etc. The study is useful for assessing sustainability, but there are several scientific weaknesses of the current manuscript, which are described in details below:

(1) Lines 64-85: The description of the theory within these lines is rather long, and has to be simplified and concise.

(2)  Lines 101-105: Should mention and highlight more on the achievements of land use change detection / identification via machine learning techniques, and existing experience or success of adopting different ML techniques in land use retrieval, for example the following references:

https://www.mdpi.com/2072-4292/13/16/3337

https://www.tandfonline.com/doi/abs/10.1080/10106049.2021.1923827

https://www.mdpi.com/2072-4292/14/5/1189

https://www.sciencedirect.com/science/article/pii/B9780323898614000221

(3) Lines 117-119: Here, the authors mentioned the key achievement of this manuscript, however the goal and vision is not clear. Is this paper focusing on techniques? Or social perspectives and analysis? Or for satisfying goals of sustainable development within Chinese community?

(4) Line 148: The authors mentioned that random forest method is used to measure the contribution of relevant driving factors. How is it being used and applied? It seems that the entire manuscript didn't mention and provide a theoretical basis of RF method.

(5) Line 161: "standardized processing" - Is there any criteria of data selection? Why are some datasets being eliminated and removed, from remote sensing perspectives / temporal reasons / social perspectives?

(6) The land use maps obtained in Figure 3: Has the use of MODIS data been well justified? Especially with respect to land use retrieval in Nanjing / nearby cities of China?

(7) Section 2.5: The implications and importance of all these formula shown here are missing. The authors should explain clearly why these formula were adopted? How these results could help with spatial analyses etc.?

(8) Lines 254-261: The authors have not provided the details of the MEAS model as they mentioned. Please add a paragraph for providing fine details.

(9) Lines 260-261: What are the key purposes of obtaining these time series?

(10) Lines 347-356: Please add some numerical quantities here, so that the presentation of results look more scientific.

(11) Table 4: What is an acceptable value here? It is clear that 0.17 > 0.03, but what is the key implication of these accuracy values?

(12) Lines 379-385: The index system looks good, but the effectiveness of usage of these long-term spatial and remote sensing data is highly doubted. Most datasets may not be available in continuous basis, so how could the authors ensure that the index system developed here can work in other applications / cities?

(13) Lines 424-434: It would be much better if some statistics / numerical values can be provided here.

(14) Lines 424-443: Can the authors convert all these suggestions and appropriate spatial planning strategies obtained to a map? i.e., add a figure with spatial representation of graphical results, with proper labelling within the map.

(15) Lines 471-475: How could the discussion be proceeded / implemented in different perspectives? For example, citizen-based, society-wise, policy-wise approaches?

Minor Modifications

Line 26, 74-79: Should describe the kind and the type of urban expansion that occurred in Nanjing, throughout different places of this manuscript - there are 3 manners, namely infilling, edge expansion and leapfrog expansion, and should compare with the type of urban expansion that took place in other cities. They should also take reference of the literature mentioned in Point (2) above.

Line 26: "Ecological land can mitigate spatial conflicts" - why? some explanations have to be provided in the manuscript.

Line 52: Capitalization should be adopted for Sensing, Information and Systems

Line 98: "multiple methods" - should provide some examples for clarity.

Figure 1: The labelling here could be clearer, it's hard to see the exact wording.

Table 2: The authors should rescale the table here, and adjust the format of this table if necessary. Currently, it contains 3 pages, which is quite messy in terms of formatting. Further, for "-" and "+", what measures do the authors mean?

Eq. (2): Why the ln function is adopted?

Lines 278-280: Proper labelling of these respective area / cities should be added to Figure 4.

Figure 4: The word "Legend" should be replaced by a certain quantity / index?

Figure 5: What do the colors here indicate? 

Lines 332-333: How does the local government deal with the use of these land areas? For natural sceneries? Any proper official documents for that?

Lines 492-496: The results obtained here within this manuscript are only limited to particular time period / years, but may not be true in general. Some proper address should be made in different parts of this manuscript.

Grammatical mistakes were found and detected throughout the manuscript, therefore, a proper round of grammatical editing should be conducted before resubmission.

Line 60: arise from

Line 124: Situated at

Line 125, add a "comma" after climate

Line 170: overlapping

Line 181: We used the "weighted" average

Line 191: expressed as

Line 194: to reflect

Line 213: overlapping

Lines 219-222: The sentence should be rewritten.

Line 233: environmental protection

Line 313: has significantly increased

Line 458: "So then" should not appear

Lines 470-471: this sentence has to be rewritten

Line 486: urban areas, and two large

Generally speaking, the authors should address all aforementioned issues and suggestions, and formulate the manuscript in a more scientific and rigorous manner, before formally resubmitting it.

Author Response

(The authors gave the same response as above.)

Round 2

Reviewer 1 Report

Thank you to the authors for addressing all suggested revisions. The definitions are better clarified.

Author Response

Thanks very much for your kind work and consideration on publication of our paper. On behalf of my co-authors, we would like to express our great appreciation to editor and reviewers.
Thank you and best regards.

Reviewer 2 Report

The authors improved the paper and I think that can be published in present form.

Author Response

(The authors gave the same response as above.)

Reviewer 3 Report

Dear authors,

The revised version of this manuscript looks much clearer and informative as a whole. There are still several places to modify / update before publication:

(1) Within the review report, the authors have mentioned different technical and data analytic shortcomings - please include these shortcoming explicitly in Section 4.5 (Discussion). Please also try to associate these shortcoming with future goals within this field.

(2) For the previous question on Random Forest Theory, we noticed that the authors have added some descriptions in different parts of this manuscript, however more information with regard to its theoretical framework should be added, like equations, the application of RF or other ML techniques in similar studies previously.

(3) The labelling within the spatial maps of Figures 5 and 7 are rather small, and can be enlarged in terms of their font size.

(4) The description of different colors in Figure 6 can be added in the caption of Figure 6.

Other than that, I think the topic is of practical interests, and the results obtained are meaningful to some extent. 
